# Modelling the supply and need for health professionals for primary health care in Ghana: Implications for health professions education and employment planning

James Avoka Asamani[1,2], Christmal Dela Christmals[1]*, Gerda Marie Reitsma[1]

1 Centre for Health Professions Education, Faculty of Health Sciences, North-West University, Potchefstroom, South Africa, 2 World Health Organization, Regional Office for Africa, Inter-Country Support Team for Eastern and Southern Africa, Harare, Zimbabwe

* christmal.christmals@nwu.ac.za

## Abstract

**Data Availability Statement:** All relevant data are within the manuscript and its Supporting Information files.

### Background

The health workforce (HWF) is critical in developing responsive health systems to address population health needs and respond to health emergencies, but defective planning have arguably resulted in underinvestment in health professions education and decent employment. Primary Health Care (PHC) has been the anchor of Ghana's health system. As Ghana's population increases and the disease burden doubles, it is imperative to estimate the potential supply and need for health professionals; and the level of investment in health professions education and employment that will be necessary to avert any mismatches.

### Methods

Using a need-based health workforce planning framework, we triangulated data from multiple sources and systematically applied a previously published Microsoft® Excel-based model to conduct a fifteen-year projection of the HWF supply, needs, gaps and training requirements in the context of primary health care in Ghana.

### Results

The projections show that based on the population (size and demographics), disease burden, the package of health services and the professional standards for delivering those services, Ghana needed about 221,593 health professionals across eleven categories in primary health care in 2020. At a rate of change between 3.2% and 10.7% (average: 5.5%) per annum, the aggregate need for health professionals is likely to reach 495,273 by 2035. By comparison, the current (2020) stock is estimated to grow from 148,390 to about 333,770 by 2035 at an average growth rate of 5.6%. The health professional's stock is projected to meet 67% of the need but with huge supply imbalances. Specifically, the supply of six out of the 11 health professionals (~54.5%) cannot meet even 50% of the needs by 2035, but Midwives could potentially be overproduced by 32% in 2030.

**Funding:** The author(s) received no specific funding for this work.

**Competing interests:** The authors have declared that no competing interests exist.

## Conclusion

Future health workforce strategy should endeavour to increase the intake of Pharmacy Technicians by more than seven-fold; General Practitioners by 110%; Registered general Nurses by 55% whilst Midwives scaled down by 15%. About US$ 480.39 million investment is required in health professions education to correct the need versus supply mismatches. By 2035, US$ 2.374 billion must be planned for the employment of those that would have to be trained to fill the need-based shortages and for sustaining the employment of those currently available.

## Introduction

Over the years, most health workforce planning has been based on either population ratio approaches or currently observed health service utilisation [1, 2]. However, in a context where the population still faces an unmet need for health services or significant disparities in the need for health services, these approaches have been found to be of limited value [3–5]. Consequently, in 2016, the World Health Assembly (WHA) adopted resolution WHA69.19 (The Global Strategy on Health Workforce), urging countries to make a paradigm shift in health workforce planning towards the use of population health needs as the basis for health workforce planning rather than the use of currently observed levels of health service utilisation, service targets, health facilities or simple population ratios. In furtherance of this, it called for health workforce investments to be based on matching ". . .the supply of health workers to population needs, now and in the future" [6]. This paradigm shift has been deemed necessary to uphold the tenets of Universal Health Coverage (UHC) to ensure that all persons have access to the health workers they will require based on their health needs and not based on their location and ability to pay gender, race or other characteristics. Thus, it has become critical for countries to devise effective policies that respond to population needs and effectively plan the future training of health professionals by quantifying the needed health workforce based on the population health needs and their supply capacity based on the evidence [6].

Ghana faces a double burden of disease whereby non-communicable diseases and their risk factors are at alarming levels, whilst communicable diseases are still a public health threat [7, 8]. With a UHC score estimated at 47% in 2019, Ghana does not only sub-optimally compares with Africa's average of 48% [9], but also has up to 53% of its population health needs (which are tracked by UHC tracer indicators) likely not to be met by the existing coverage of health services. Addressing the aforesaid and ever-changing pattern of the population's health needs require investments across the different health system components, but critically aligning the health workforce production to the population health needs is imperative. Like many other low and middle-income countries, Ghana has faced a critical shortage of health workers, undermining health service coverage [10–13].

To address the severe health workforce shortages in the late 1990s, Ghana expanded its public and private-sector production of the health workforce, resulting in increasing the public sector health workforce by nearly three-fold between 2005 and 2019 [14], following which the country is being cited as one of the leading producers of physicians, nurses and midwives in sub-Saharan Africa [15]. Nonetheless, based on national health facility staffing norms [16], Ghana is estimated to have at least a 42% shortfall of the health professionals needed [13]. On the one hand, the shortfall is more critical amongst highly trained health professionals, while

on the other hand, some mid-level health workers appear to have been overproduced [13]. It is also worth noting that these challenges co-exist with several trained but unemployed health workers in the country, mainly due to inadequate investments in health sector employment [14]. Recent estimates show that there is a need for a public sector health workforce budgetary increase of 57% (~US$295.4 million) to meet minimum staffing requirements for primary and secondary level health services [17]. Such demands are becoming difficult to justify as some 22% (range: 14%– 50%) of the prevailing wage bill were said to have been spent on health workers who were considered inequitably distributed [17], which tended to favour specialised health facilities in large cities [11, 18]. This has, thus, left significant staffing gaps at the primary health care (PHC) level, where 95% of all outpatient services were provided to the majority of the population [19]. Therefore, several policy questions continue to persist. These questions include, amongst others, how many health professionals are needed at the PHC level based on the population health needs viz-a-viz potential supply and what level of investment in health professions education and employment will be required to avert any shortages?

To address these critical policy questions, we employed a population needs-based approach to model the requirements of eleven categories of health professionals at the primary health care level in Ghana, alongside their supply projections based on a stock-and-flow approach. Together, these health categories of professionals constitute more than 80% of the total wage bill and are the main anchor of primary health care interventions.

## The primary healthcare context in Ghana

Ghana is politically and administratively divided into sixteen (16) regions, further divided into 260 districts. Each district is also divided into health sub-districts and communities in which the delivery of health services is operationally and administratively aligned with these structures. By a legislative act (Act 525 of 1996), the public health service delivery is entrusted in semi-autonomous establishments, namely the Ghana Health Service (GHS) and Teachings Hospitals (THs). There are also several faith-based, private-for-profit and quasi-governmental institutions and mental health facilities. The Ministry of Health (MOH) supervises all these institutions focuses on policy formulation, resource mobilisation and allocation, and monitoring and evaluation.

As indicated in Fig 1, a pluralistic gatekeeper health service delivery system is established. At the bottom of the primary health care hierarchy are community-based health planning and services (CHPS), the first point of contact of the health system, each serving a defined

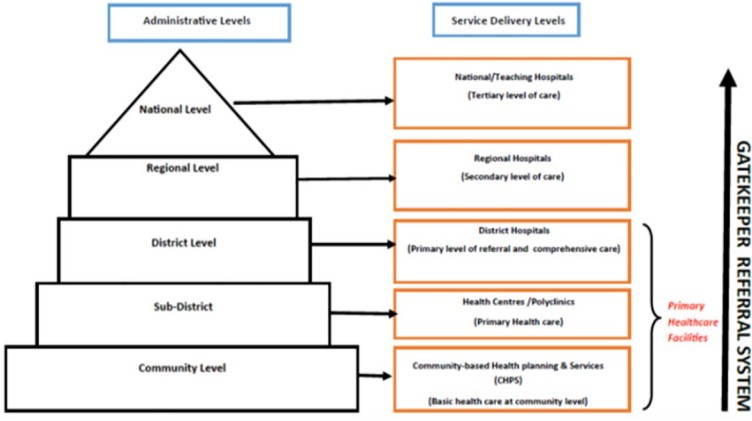

**Fig 1. The health care system structure in Ghana [Source: Authors' construction].**

population of 5,000 people or 750 households, sometimes coterminous with electoral areas [20]. These are mandated to deliver preventive health services and treat minor illnesses with over-the-counter medicines at the community level.

Health centres (HCs) serve people in the catchment area of a sub-district and are expected to provide basic curative and preventive health services. They are the first level of referral from the communities and CHPS. Although health centres are mandated to serve catchment populations of about 20,000 or less, in urban areas, they could be expanded (in size and staffing) and designated as polyclinics to serve larger populations. One report revealed that some urban polyclinics were even operating at the level of hospitals, beyond their designation, due to increased service demand from the population [17]. At the top of the primary health care hierarchy are the district/primary hospitals intended to serve as district-level referral centres and provide preventive, curative, and emergency health care to populations between 100,000 and 200,000. However, it is estimated that at least half of the districts were without these primary/district hospitals, resulting in extra workload in other facilities, which operate beyond their capacity [11, 17].

## Methodology

### Overview of population needs-based simulation model for health workforce planning

Based on a scoping review of various analytical applications of needs-based health workforce planning approach [21], and building upon previous works [4, 22–26], we developed a needs-based analytical health workforce planning model, built in Microsoft® Excel and suitable for health sector-wide application in any country [27]. The paper reports on how the model was applied to conduct health workforce projection in PHC in Ghana. As illustrated in Fig 2, the

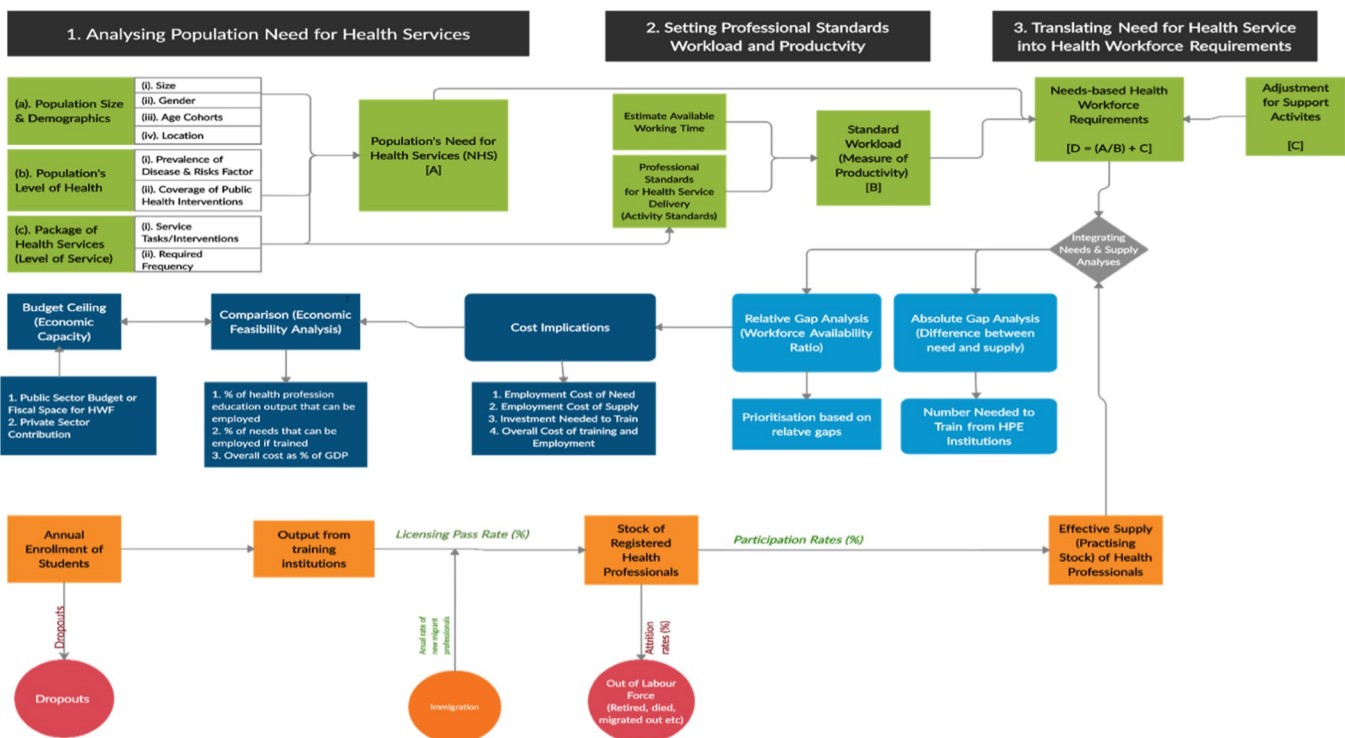

**Fig 2. A conceptual framework for the population needs-based simulation model for health workforce planning.** Source: Adopted from Asamani et al. [27].

underpinning conceptual framework of the model consists of need analysis, supply analysis, gap analysis, and resource implications.

The need analysis assumes that the required calibre and quantity of health professionals are derived from the population's need for health services [2, 5, 28, 29]; as the population in any jurisdiction at any given time have some need for health services, regardless of whether or not they have demanded such service or if it can even be afforded [2]. Therefore, forecasting the optimal requirement for health professionals in a health system must directly flow from an estimate of the population's need for health services which can be modelled as a function of (a) the size of the population and its demographic characteristics (b) the state of health or level of health of the population and (c) the level of services (type and frequency) necessary to attain and maintain optimal health of the population. The aggregate need for health services can then be translated into the health workforce requirements if the category of health professional competent to deliver the service was identified [24] with clear work division [26], and an established measure of their standard workload (or productivity) [4, 25]. Box 1 summarises the empirical formulae for estimating the need for health services and translating the same into health workforce requirements. See S1 File for the fully functional excel-based model.

Building on existing stock and distribution of the health workforce, the model uses the stock and flow approach to estimate health professionals' future supply. It comprises determining the inflow or entry in the current workforce and the outflow or attrition from the current workforce. While the inflow depends on the training capacity and immigration, the outflow/attrition is influenced by natural and unnatural disengagements such as retirements, emigration, deaths, resignations and dismissals. Box 2 presents the supply projection formula, explained in detail in a prior publication [27].

The gap analysis compares the need analysis and the supply analysis to determine if there were a potential need-based shortage or oversupply of the health professionals [30]. Absolute gaps are presented as the actual number of deficit or surplus of health professionals, whilst relative gaps were presented as the proportion of the need-based requirements that the anticipated supply levels could potentially meet. While absolute gaps are essential for planning the number of additional health professionals to train, relative gaps are significant in prioritising the health professional group that required immediate attention. It also has implications for quality of care as it could be interpreted to mean the degree to which professionals standards will be met [31]. Finally, the resource implications are computed in terms of the investment required to sustain jobs for the anticipated supply can be compared with the investment needed to fill the need-based requirement assuming there were no supply-side barriers (see Box 3 for formulae). Where there are need-based shortages, the required investment in health professions education to fill the gaps are estimated. All these cost implications are then compared with budgetary allocations or potential financial space to examine their feasibility.

## The empirical application of the model: Data sources and assumptions

Input data for the model application (Eqs 1–8) were triangulated from multiple sources. This section highlights the nature and process of data acquisition and data validation procedure and limitations.

## Data sources for the need analysis

**Population size and characteristics.** The population size, gender, age composition (age cohorts) and their distribution by geographical regions and rural and urban distribution were taken from the Ghana Statistical Service projections [32].

Box 1. Summary of formulae for estimating needs-based health workforce requirements.

$$NHS_t = \sum P_{i,j,t,g} \times [H_{h,i,j,t-1} \times (1 + R_h)] \times L_{y,h,i,j,t} \qquad \text{Eq 1}$$

Where:

- $NHS_t$ represents the 'Needed Health Services' by a given population under a given service model, $L_{i,j,t}$ over a period of time $t$.

- $P_{i,j,g,t}$ represents the size of the given population of age cohort $i$, gender $j$ in location (rural or urban) $g$ at time $t$ in a given jurisdiction (this represents the population and its demographic characteristics).

- $H_{h,i,j,g,t}$ represents the proportion of the given population with health status $h$, of age cohort $i$, gender $j$ in location $g$ at time $t$ (this represents the level of health of the population).

- $L_{y,h,i,j,g,t}$ represents the frequency of health services of type $y$ planned or otherwise required, under a specified service model, to address the needs of individuals of health status $h$ among age cohort $i$, gender $j$ in location $g$ over time $t$ (this represents the level of service required by the population).

- $R_h$ is the instantaneous rate of change of the health status, $h$.

$$SW_{n,y} = \frac{AWT_n}{SS_{y,n}} \qquad \text{Eq 2}$$

Where:

- $SW_{n,y}$ is the standard workload for health professionals of category $n$ when performing health service activity $y$.

- $AWT_n$ is the annual available working time of the health professional of category $n$

- $SS_{y,n}$ is the Service Standard or the time it takes a well-trained health professional of category $n$ to deliver the service activity, $y$.

$$N_{n,t} = \Sigma \frac{NHS_{n,y,t}}{SW_{n,y}} \qquad \text{Eq 3}$$

Where:

- $N_{n,t}$ is the need-based requirements of a health professional of category $n$ at time point $t$

- $NHS_t$ represents the number of needed health service activity $y$, delivered by a health professional of category $n$ at time $t$.

- $SW_{n,y}$ is the standard workload for health professionals of category $n$ when performing health service activity $y$

Sources: Adopted from Asamani et al. [27].

> ## Box 2. Stock and flow formulae for health professionals' supply estimation
>
> $$S_{n,t} = [T_{n,t-1} \times (1 - a_n) + I_n] \times P \qquad \text{Eq 4}$$
>
> Where:
>
> - $S_{n,t}$ is the effective supply of health professionals of category $n$, at time $t$.
>
> - $T_{n,t}$ is the overall stock of health professionals (number registered) category $n$ at time $t$.
>
> - $a_n$ is the rate of attrition (the proportion of the stock, $T_{n,t-1}$ that died, retired, could not work due to ill-health or migrated out) which adjusts the overall stock to get the professionally active health professional of category $n$.
>
> - $I_n$ is the inflows of the health professional of category $n$ from both domestic and foreign sources.
>
> - $P$ is the rate of labour participation that reflects the proportion of the professionally active health professionals engaged in direct health service delivery.
>
> Sources: Adopted from Asamani et al. [27].

**Level of health (disease burden).** The list of diseases and risk factors that accounted for 98% of the burden of mortalities, outpatient attendance and hospital admissions were profiled from the Global Burden of Disease Study [33] and the Facts and Figures report of the Health Sector in Ghana [34, 35]. We then conducted a desk review of survey reports, analytical reports and peer-reviewed publications to map the prevalence or incidence (as appropriate) of the diseases and risk factors identified and the coverage rates of essential public health interventions. The notable sources of the data included the Ghana Demographic and Health Surveys [36, 37], Ghana Maternal Health Surveys [38, 39], the Facts and Figures of the health sector in Ghana [34, 35], Ministry of Health annual holistic assessment report of the health sector [19, 40, 41], and several peer-reviewed publications (see S1 Table for the details of diseases and risk factors with their prevalence or incidence rate as well as the data sources).

**Level of service.** The main health services that were being provided or were otherwise necessary at the PHC level to address the diseases and risk factors identified above were extracted from various sources, including the Ghana Standard Treatment Guidelines 2017 [42]; CHPS policy [20]; National Reproductive Health Service Policy and Standards [43]; the Non-Communicable Disease Strategy [44]; Ghana National HIV and AIDS Strategic Plan [45]; and clinical management guidelines for TB and HIV infection [46]. The health service activities were matched to the job description of the health professionals being studied.

**Activity standards and standard workloads.** The list of tasks performed by health professionals (matched with the services required to address the disease burden identified above) was extracted from the job descriptions of the respective health professionals and those of a previous Workload Indicators of Staffing Needs (WISN) study [47]. The standard time it would take the individual health professionals to perform these health service tasks were elicited through a nationally representative cross-sectional survey completed by 503 health professionals (the detailed methodology and results of the survey have been reported in a separate piece) [48].

## Box 3. Formulae for gap analysis and cost implications

$$Absolute\ Gap_{n,t} = S_{n,t} - N_{n,t} \qquad \text{Eq 5}$$

$$SAR_{n,t} = \frac{S_{n,t}}{N_{n,t}} \qquad \text{Eq 6}$$

- *Absolute Gap*$_{n,t}$ is the absolute gap for health professionals of type *n* at time *t*.
- $S_{n,t}$ is the supply of health professionals of category *n* at time *t*.
- $N_{n,t}$ is the needs-based requirements of a health professional of category *n* at time *t*.

$$TCS_{n,t} = \Sigma(S_{n,t} \times K_{n,t}) \qquad \text{Eq 7}$$

Where:

- $TCS_{n,t}$ is the total wage bill cost of the anticipated supply of health professional category *n* at time point *t*.
- $S_{n,t}$ is the anticipated supply of health worker category *n* at time point *t*.
- $K_{n,t}$ is the average income (made up of salaries, allowances and monetary benefits and adjusted for inflation) for health professionals of category *n* at time point *t*.

$$TCN_{n,t} = \Sigma(N_{n,t} \times K_{n,t}) \qquad \text{Eq 8}$$

- $TCN_{n,t}$ is the total wage bill cost of need-based requirements of a health professional of category *n* at time point *t*.
- $N_{n,t}$ is the need-based requirements of a health professional of category *n* at time point *t*.
- $K_{n,t}$ is the average income (made up of salaries, allowances and monetary benefits and adjusted for inflation) for health professionals of category *n* at time point *t*.

Source: Adopted from Asamani et al. [27].

**Workload division.** We adopted a workload division established by the Technical Working Group (TWG) on Staffing Norms of the Ministry of Health in 2014 [16, 47]. The TWG adopted a workload division of 25% of medical laboratory workload for the Biomedical Scientist and 75% for Laboratory Technicians, noting that most laboratory examinations, especially at the PHC level, were not complex and hence the role of the Biomedical Scientist was for quality assurance undertaking the 25% of the workload which may require a higher level of technical knowledge or skill. Based on similar logic, pharmacy-related workload division of 20% for Pharmacists and 80% for Pharmacy Technicians was also adopted by the TWG. Based on observed data, the TWG also concluded that Physician Assistants covered 20% of the workload

in terms of outpatient consultations in primary/district hospitals and polyclinics, leaving 80% for General Practitioners. However, the Physician Assistant covered 80% of outpatient consultations at health centres, with the remaining 20% taken care of by nurses. We also adopted the workload ratio of 70% for professional nurses (Registered General Nurses) and 30% for auxiliary nurses (Enrolled Nurses) for clinical nursing care. For Midwives, Community Health Nurses, Nutritionists and Dieticians, their functions at the PHC level are usually not shared with other professionals; hence, we made no workload division assumptions.

Data from the health sector holistic assessment by the MOH show that between 2014 and 2017, on average, about 20% of outpatient consultations were provided by health professionals at CHPS compounds/zones; 26% at health centres/polyclinics; 49% at primary (and district) hospitals while the rest of 5% was provided in either secondary or tertiary health facilities [19]. We applied the above-mentioned observed trends to the modelled need for health services for the division of service delivery at the various levels of PHC.

## Data sources for the supply analysis and costing

The existing stock of health professionals, the rate of labour flow (attrition), the education pipeline (number of admissions into health professions education institutions and pass rates) were obtained from the respective professional regulatory bodies of the health professions (as indicated in Table 1). The health professionals' average income level was taken from the public sector single spine salary scale obtained from the Ghana Health Service. In the absence of comprehensive data on the cost of training of health professionals, we used the average of annual fees paid by fee-paying students as published by two public universities (the University of Ghana's College of Health Sciences and the University of Health and Allied Sciences) and one private university (the Central University). Since fee-paying students in public and private universities do not benefit from government subsidies and are usually charged at least for full cost recovery for tuition and other costs of training, we assumed that it better reflected the 'true cost' of training as compared with the regular student fees which the government substantially subsidizes. However, the estimated cost of training per student per year excluded boarding and lodging, which enormously varies depending on the cities and lifestyles of individual students.

## Data validation and quality assurance

Data extracted from official reports and websites of MOH, GHS, and the respective health professions regulatory bodies were sent to focal persons in the respective institutions to confirm the validity of the data and explanations was provided for any inconsistencies observed. They also indicated whether or not any subsequent update to the data or report was made and available. Where there was unexplained data inconsistency, a comparison was made with international datasets (if available) such as the World Development Indicators of the World Bank, WHO's Global Health Observatory (GHO), and the National Health Workforce Account (NHWA) database. Data obtained from peer-reviewed publications were also compared with papers of similar methodology to ensure consistency of estimates. Whenever there was wide variation in estimation between two publications, additional papers were sought for further comparison, and the closest estimates were used. Two of the authors systematically and consistently scrutinised the data before analysis.

## Ethical considerations

This study did not involve the use of human subjects as it was based on publicly available data and documents. However, as larger project, the study received ethics approval by the Health

**Table 1. Baseline stock, labour flows, training outputs, average income levels and cost of training.**

| No. | List of cadres | Active Baseline Stock | Rate of Annual Retirements/ Death (%) | Rate of other forms of attrition | Overall Annual Attrition Rate (%) | Duration of training (Years) | Average Number of Annual Enrolments | Average Pass Rate (%) | Average Annual Income (GH¢) | Cost of Training Per Year (GH¢) | Sources of Data |
|---|---|---|---|---|---|---|---|---|---|---|---|
| 1 | General Practitioner (Generalist Doctor) | 6,173 | 1.9% | 5.2% | 7.1% | 6 | 1,566 | 70.0% | 68,182.3 | 8,242.0 | MDC; GHS; MOH; UG; UHAS |
| 2 | Physician Assistant (Medical) | 3,118 | 1.9% | 0.0% | 1.9% | 3 | 638 | 80.5% | 35,821.4 | 10,712.5 | MDC; GHS; MOH; UG; CU |
| 3 | Midwife | 12,786 | 7.6% | 0.3% | 7.9% | 3 | 4,827 | 79.0% | 28,290.1 | 8,910.5 | NMC; GHS; MOH; UHAS; UG |
| 4 | Registered General Nurse | 60,530 | 2.8% | 0.4% | 3.2% | 3 | 7,353 | 73.0% | 28,290.1 | 8,910.5 | NMC; GHS; MOH; UHAS; UG; CU |
| 5 | Enrolled Nurse | 40,000 | 1.6% | 3.9% | 5.5% | 2 | 8,379 | 86.0% | 18,260.4 | 8,910.5 | NMC; GHS; MOH; UHAS |
| 6 | Community Health Nurse | 24,494 | 0.8% | 3.9% | 4.7% | 2 | 4,184 | 92.0% | 18,260.4 | 8,910.5 | NMC; GHS; MOH; UHAS |
| 7 | Nutritionist and Dietician | 334 | 2.1% | 0.0% | 2.1% | 3 | 429 | 92.8% | 35,821.4 | 8,089.0 | AHPC; GHS; MOH; UHAS; UG |
| 8 | Biomedical Scientist* | 1,355 | 2% | 0.0% | 2% | 4 | 396 | 92.8% | 35,821.4 | 8,089.0 | AHPC; GHS; MOH; UHAS; UG |
| 9 | Laboratory Technician | 855 | 2.1% | 0.0% | 2.1% | 3 | 682 | 92.8% | 28,290.1 | 8,089.0 | AHPC; GHS; MOH; UHAS; UG |
| 10 | Pharmacist** | 1,052 | 0.8% | 0.1% | 0.9% | 4 | 343 | 81.6% | 38,317.4 | 7,594.0 | PCG; GHS; MOH; UHAS; UG |
| 11 | Pharmacy Technician | 1,055 | 2.1% | 0.0% | 2.1% | 3 | 173 | 81.6% | 26,007.8 | 7,594.0 | PCG; GHS; MOH; UHAS; UG |

*a six-year Doctor of Medical Laboratory programme has been introduced by some universities

**a six-year Doctor of Pharmacy programme is being introduced by universities, but the 4-year Bachelor of Pharmacy degree is still the basic requirement to be a Pharmacist in the country.

Other forms of attrition included resignations, vacation of posts and dismissals.

AHPC–Allied Health Professions Council; CU–Central University; GHS–Ghana health Service; MDC–Medical and Dental Council; MOH–Ministry of Health; NMC–Nursing and Midwifery Council; PCG–Pharmacy Council of Ghana; UG–University of Ghana; UHAS–the University of Health and Allied Sciences

Research Ethics Committee (HREC) of North-West University in South Africa with number NWU-00416-20-A1 and the Ghana Health Service Ethics Committee with number GHS-ER17/07/20. Access to administrative datasets was approved by the Director-General of the Ghana Health Service.

## Findings

### Projected supply of health professionals, 2020–2035

In aggregate, the current stock of 11 health professionals considered in this analysis was estimated to be 148,390 across public and private sectors, with the projection showing a progressive increase over the next 15 years. Overall, the average net annual growth rate is estimated to be 5.6% resulting in a rise of 51.9% from the current stock to 225,454 by 2025 and a further 26.8% increase to 285,900 by 2030, which could reach 333,770 by 2035. Thus, by 2035, holding the current rate of production and attrition constant, the size of the 11 categories of health professionals considered in this analysis is likely to be 2.5-fold that of the bassline stock in 2020.

General Practitioners stock is expected to increase by 46.2% from 6,173 in 2020 to 9,027 by 2025, with a further increase of 12.19% by 2030 and capping at about 12,369 by 2035 (an increase of 12.4% from the stock expected in 2030). Similarly, the supply of Physician Assistants is anticipated to increase by almost 70% from 3,118 in 2020 to 5,305 in 2025 and 37.5% and 24.8% by 2030 and 2035, respectively. Also, with the double streams of producing Midwives via direct entry and post-basic training, the stock of Midwives is likely to almost double (93.6% increase) from 12,786 in 2020 to 24,756 in 2025, and a further expansion of 53.3% could be reached by 2030 if the prevailing rate of production and attrition were held constant.

Registered General Nurses' active stock is projected to increase by 27% from 59,986 in 2020 to 76,158 by 2025, with a further 18% increase to 89,903 by 2030. If the production rate continued unabated, the stock of Registered General Nurses could reach 101,585 by 2035. Besides, the baseline stock of 24,494 Community Health Nurses is projected to reach 54,108 (120.9% increase) by 2035 as were Enrolled Nurses to increase from 37,182 in 2020 to 77,816 by 2030 (109% increase) and 90,998 by 2035 or 16.9% further boost from the stock anticipated in 2030. Finally, the expected output of Laboratory Technicians showed a ramp-up in recent years whereby the baseline stock of 855 is likely to increase by 3.5-fold to 3810 by 2025 and nearly 10-fold to 8,857 by 2035. Table 2 provides details of the annual projections of health professionals' supply if the current trend continued without interventions to either abate or accelerate the production.

### Projected need-based requirements for health professionals, 2020–2035

The projections show that based on the population (size and demographics), disease burden, the package of health services and the professional standards for delivering those services, Ghana needed 221,593 health professionals across the 11 categories included in the analysis for primary health services. At the rate of change of 5.5% (range: 3.2% - 10.7%) per annum, the aggregate requirement is likely to reach 407,897 by 2030 (84.1% increase) and a further 21.4% increase to 495,273 by 2035.

Specifically, General Practitioners need-based requirement is estimated to be roughly 14,049 in 2020, which is anticipated to averagely increase at an annual rate of 5.7% (range is 2.7% to 11.2%). With this trajectory, the need for General Practitioners could increase by 89% to 26,560 by 2030 and then reach 32,199 by 2035, an additional needs-based increase of 21.2% between 2030 and 2035.

Physician Assistants baseline requirement is estimated to be 8,590 in 2020 and is expected to double to 17,633 (an increase of 105%) by 2030 and almost 21,487 by 2035, about 2.5-fold that of the baseline requirement in 2020. Also, about 14,002 Midwives are estimated to be

**Table 2. Projected supply of health professionals, 2020–2035.**

| No. | Health Professionals | Projected Supply, 2020–2035 | | | | | | | | | | | | | | | |
|---|---|---|---|---|---|---|---|---|---|---|---|---|---|---|---|---|---|
| | | 2020 | 2021 | 2022 | 2023 | 2024 | 2025 | 2026 | 2027 | 2028 | 2029 | 2030 | 2031 | 2032 | 2033 | 2034 | 2035 |
| 1. | Community Health Nurse | 24,494 | 27,197 | 29,773 | 32,229 | 34,570 | 36,802 | 38,929 | 40,956 | 42,889 | 44,731 | 46,487 | 48,160 | 49,756 | 51,276 | 52,726 | 54,108 |
| 2. | Enrolled Nurse | 37,182 | 42,350 | 47,236 | 51,853 | 56,217 | 60,343 | 64,242 | 67,927 | 71,411 | 74,703 | 77,816 | 80,757 | 83,538 | 86,166 | 88,650 | 90,998 |
| 3. | General Practitioner (Generalist Doctor) | 6,173 | 6,831 | 7,442 | 8,010 | 8,537 | 9,027 | 9,483 | 9,906 | 10,299 | 10,664 | 11,003 | 11,318 | 11,610 | 11,882 | 12,135 | 12,369 |
| 4. | Laboratory Scientist | 1,355 | 1,696 | 2,029 | 2,356 | 2,677 | 2,991 | 3,299 | 3,600 | 3,896 | 4,186 | 4,470 | 4,748 | 5,021 | 5,288 | 5,550 | 5,806 |
| 5. | Laboratory Technician | 855 | 1,471 | 2,075 | 2,665 | 3,244 | 3,810 | 4,364 | 4,907 | 5,438 | 5,958 | 6,467 | 6,966 | 7,454 | 7,931 | 8,399 | 8,857 |
| 6. | Midwife | 12,786 | 15,589 | 18,171 | 20,549 | 22,739 | 24,756 | 26,613 | 28,324 | 29,900 | 31,351 | 32,688 | 33,919 | 35,053 | 36,097 | 37,058 | 37,944 |
| 7. | Nutritionist and Dietician | 334 | 735 | 1,127 | 1,511 | 1,886 | 2,254 | 2,615 | 2,967 | 3,312 | 3,650 | 3,981 | 4,305 | 4,622 | 4,933 | 5,237 | 5,534 |
| 8. | Pharmacist | 1,052 | 1,322 | 1,590 | 1,856 | 2,119 | 2,380 | 2,638 | 2,895 | 3,148 | 3,400 | 3,649 | 3,896 | 4,141 | 4,384 | 4,624 | 4,862 |
| 9. | Pharmacy Technician | 1,055 | 1,175 | 1,292 | 1,407 | 1,519 | 1,629 | 1,737 | 1,842 | 1,945 | 2,046 | 2,145 | 2,242 | 2,337 | 2,429 | 2,520 | 2,609 |
| 10. | Physician Assistant (Medical) | 3,118 | 3,572 | 4,018 | 4,455 | 4,884 | 5,305 | 5,718 | 6,123 | 6,520 | 6,910 | 7,292 | 7,667 | 8,035 | 8,396 | 8,750 | 9,097 |
| 11. | Registered General Nurse | 59,986 | 63,434 | 66,772 | 70,003 | 73,131 | 76,158 | 79,089 | 81,926 | 84,672 | 87,330 | 89,903 | 92,394 | 94,805 | 97,139 | 99,398 | 101,585 |
| | **Total** | 148,390 | 165,372 | 181,525 | 196,894 | 211,524 | 225,454 | 238,725 | 251,372 | 263,429 | 274,929 | 285,900 | 296,372 | 306,370 | 315,921 | 325,047 | 333,770 |
| | Net Annual Increase | | 16,982 | 16,153 | 15,369 | 14,629 | 13,931 | 13,271 | 12,647 | 12,057 | 11,499 | 10,971 | 10,472 | 9,999 | 9,551 | 9,126 | 8,724 |
| | Annual % net increase | | 11.4% | 9.8% | 8.5% | 7.4% | 6.6% | 5.9% | 5.3% | 4.8% | 4.4% | 4.0% | 3.7% | 3.4% | 3.1% | 2.9% | 2.7% |

Note: Supply values for 2020 are not projections but the baseline data

needed in 2020, increasing progressively at an annual average rate of 4.9% (range: 4.5% -5.5%) to 17,586 by 2025 and 28,805 by 2035 (about 106% increase between 2020 and 2035).

At an annual rate of increase averaging 5.5% (range: 3.1% - 12.5%), the need for Registered General Nurses could increase from 66,948 at baseline in 2020 to 148,983 by 2035, which represents a cumulative change of ~123% increase between 2020 and 2035. For Enrolled Nurses, the needed number in 2020 is estimated to be 45,354 and is anticipated to grow at an annual rate of 5% (range: 3.1% to 9.6%) to reach 94,381 by 2035.

Similarly, the need for Community Health Nurses is estimated to be 41,787 at baseline in 2020 and is likely to increase at an average annual rate of 5.7% (range is 3.3% to 12%) to 77,071 by 2030 and 96,233 by 2035; as the need for 5,937 Nutritionists and Dietitians is also likely to increase by 19.5% to 7,094 in 2035. The need for Biomedical scientists could also increase by 3.3-fold from 4,581 in 2020 to 14,933 by 2035, similar to Laboratory Technicians, whose estimated need is 8,585 in 2020 but projected to increase dramatically by 2.5-fold to 21,663 in 2035. Finally, the need for Pharmacist is estimated to grow at an annual average of 6.6% (range is 3.4% to 12.4%) from 3,993 in 2020 to 10,340 by 2035, similar to the Pharmacy Technicians whose need-based requirement is projected to escalate by 147% from 7,766 in 2020 to 19,154 by 2035 (an annual average growth rate of 6.3%, range: 3.1% - 11.7%). Table 3 provides the projected annual need for various health professionals.

## Health professionals' supply versus need gap analysis, 2020–2035

We compared the projected supply of the health professionals with that of the projected needs, which showed that at baseline, the stock of the health professionals included in the analysis met about 67% of their aggregate need-based requirements in 2020, leaving a gap of 33%, translating into a need-based shortage of 73,203 health professionals across 11 cadres. Without any intervention to increase the production of these health professionals, the ratio of future supply to the need (staff availability ratio) is likely to remain fairly constant (with marginal fluctuations) until 2030, when it is likely to be roughly 70%%. However, the absolute gaps will likely increase by 66.7% from 73,203 in 2020 to 121,997 by 2030 and then to 161,502 by 2035.

However, beneath the aggregate estimates are huge imbalances whereby there is a seemingly adequate production of Enrolled Nurses and an anticipated overproduction of Midwives.

**Table 3. Projected needs-based requirements of health professionals, 2020–2035.**

| No. | Health Professionals | Needs-based Requirements, 2020–2035 | | | | | | | | | | | | | | | |
|---|---|---|---|---|---|---|---|---|---|---|---|---|---|---|---|---|---|
| | | 2020 | 2021 | 2022 | 2023 | 2024 | 2025 | 2026 | 2027 | 2028 | 2029 | 2030 | 2031 | 2032 | 2033 | 2034 | 2035 |
| 1. | Community Health Nurse | 41,787 | 46,816 | 51,499 | 56,121 | 59,159 | 62,267 | 64,712 | 67,368 | 70,259 | 73,411 | 77,071 | 80,873 | 85,046 | 89,637 | 93,125 | 96,233 |
| 2. | Enrolled Nurse | 45,354 | 48,317 | 52,048 | 57,064 | 61,078 | 66,649 | 68,746 | 70,950 | 73,271 | 75,718 | 79,246 | 82,048 | 85,021 | 88,180 | 91,252 | 94,381 |
| 3. | General Practitioner (Generalist Doctor) | 14,049 | 15,366 | 16,875 | 18,765 | 20,169 | 21,923 | 22,692 | 23,520 | 24,414 | 25,383 | 26,560 | 27,712 | 28,973 | 30,358 | 31,352 | 32,199 |
| 4. | Laboratory Scientist | 4,581 | 5,043 | 5,680 | 6,592 | 7,298 | 8,151 | 8,613 | 9,131 | 9,713 | 10,369 | 11,125 | 11,966 | 12,921 | 14,009 | 14,577 | 14,933 |
| 5. | Laboratory Technician | 8,585 | 9,486 | 10,835 | 12,894 | 14,039 | 15,282 | 15,763 | 16,282 | 16,844 | 17,456 | 18,150 | 18,882 | 19,687 | 20,574 | 21,175 | 21,663 |
| 6. | Midwife | 14,002 | 14,639 | 15,312 | 16,023 | 16,775 | 17,586 | 18,429 | 19,321 | 20,265 | 21,264 | 22,440 | 23,569 | 24,766 | 26,035 | 27,379 | 28,805 |
| 7. | Nutritionist and Dietician | 5,937 | 5,931 | 5,932 | 5,941 | 5,959 | 6,038 | 6,080 | 6,132 | 6,194 | 6,266 | 6,445 | 6,548 | 6,664 | 6,793 | 6,936 | 7,094 |
| 8. | Pharmacist | 3,993 | 4,488 | 5,004 | 5,592 | 6,104 | 6,790 | 7,054 | 7,337 | 7,642 | 7,970 | 8,372 | 8,759 | 9,179 | 9,637 | 10,004 | 10,340 |
| 9. | Pharmacy Technician | 7,766 | 8,673 | 9,616 | 10,693 | 11,626 | 12,906 | 13,372 | 13,871 | 14,406 | 14,983 | 15,726 | 16,406 | 17,146 | 17,952 | 18,583 | 19,154 |
| 10. | Physician Assistant (Medical) | 8,590 | 9,775 | 10,929 | 12,157 | 13,171 | 14,511 | 15,023 | 15,578 | 16,180 | 16,837 | 17,633 | 18,422 | 19,290 | 20,247 | 20,921 | 21,487 |
| 11. | Registered General Nurse | 66,948 | 73,001 | 80,527 | 90,625 | 97,194 | 105,290 | 108,627 | 112,155 | 115,892 | 119,861 | 125,129 | 129,711 | 134,615 | 139,879 | 144,507 | 148,983 |
| | **Total** | **221,593** | **241,534** | **264,257** | **292,467** | **312,573** | **337,392** | **349,111** | **361,645** | **375,081** | **389,518** | **407,897** | **424,898** | **443,309** | **463,300** | **479,812** | **495,273** |
| | Net Annual Increase | | 19,942 | 22,722 | 28,210 | 20,106 | 24,819 | 11,719 | 12,534 | 13,436 | 14,437 | 18,379 | 17,001 | 18,411 | 19,992 | 16,512 | 15,461 |
| | Annual % net increase | | 9.0% | 9.4% | 10.7% | 6.9% | 7.9% | 3.5% | 3.6% | 3.7% | 3.8% | 4.7% | 4.2% | 4.3% | 4.5% | 3.6% | 3.2% |

Note: 2020 values are the need-based baseline requirements

At the same time, for six out of 11 (or 54.5%) health professionals considered in this analysis (namely, General Practitioner, Laboratory Scientist, Laboratory Technician, Pharmacist, Pharmacy Technician and Physician Assistant), the projected supply will likely fail to meet even 50% of the need-based requirement by 2035 if no corrective intervention(s) is undertaken to enhance health professions education.

At baseline in 2020, Midwives and Registered General Nurses appear to have marginal need-based shortages of 8.7% and 10.4%, respectively. However, over 15 years, the supply of Registered General Nurses is likely to meet only 68.2% of the need if the trajectory of need and supply remains unchanged. On the other hand, the supply of Midwives is likely to exceed that of the need by 41% in 2025, 46% by 2030 and then 32% by 2035 if the levels of production (and supply) viz-a-viz the need remain constant. Additionally, the projected supply of Enrolled Nurses met about 82% of the need-based requirement and is anticipated to incrementally improve to between 96.4% and 98.2% from 2030 to 2035 –reaching a near-equilibrium between need and supply. The needs-based shortage of Community Health Nurses is estimated to be 17,293, which represents a 41.4% need-based shortfall in 2020 which, given a high rate of internal attrition to other professions like midwifery and general nursing, the need-based shortage is likely to worsen to 44% by 2035; leaving a need for additional 42,125 Community Health Nurses in 2035.

Similarly, General Practitioners baseline stock represents only 43.9% of the need-based requirement, and the rate of supply seems to be outpaced by the rate of increasing need, leaving a need-based shortfall increasing from 7,876 in 2020 to 19,830 by 2035. Under the current trend, the need-based shortage of General Practitioners could worsen from 56.1% to 55.7% by 2030 and further escalate to a shortage of 61.3% of General Practitioners by 2035. Additionally, the baseline stock of Physician Assistants in 2020 covers only 36.3% of the need, with the shortage estimated to be 5,472, which is likely to increase steadily to 12,390 by 2035. During this time, the projected supply will represent only 42.3% of the need-based requirements.

Furthermore, Nutritionists and Dieticians are projected to have the most severe shortage at baseline, which the stock meets just 5.6% of the need; leaving a need-based shortage of 94.4%, but owing to an already started massive expansion in intake in the last few years, will likely reduce the need-based shortage to 38.2% by 2030 and 22% by 2035. This will translate into reducing the absolute shortage of Nutritionists and Dieticians from 5,603 in 2020 to only 1,560

**Table 4. Supply versus need-based gaps, 2020–2035.**

| No. | Health professionals | 2020 | | | | 2025 | | | | 2030 | | | | 2035 | | | |
|---|---|---|---|---|---|---|---|---|---|---|---|---|---|---|---|---|---|
| | | Need (a) | Supply (b) | Gap (b-a) | SAR (b/a) | Need (a) | Supply (b) | Gap (b-a) | SAR (b/a) | Need (a) | Supply (b) | Gap (b-a) | SAR (b/a) | Need (a) | Supply (b) | Gap (b-a) | SAR (b/a) |
| 1 | Community Health Nurse | 41,787 | 24,494 | (17,293) | 58.6% | 62,267 | 36,802 | (25,465) | 59.1% | 77,071 | 46,487 | (30,585) | 60.3% | 96,233 | 54,108 | (42,125) | 56.2% |
| 2 | Enrolled Nurse | 45,354 | 37,182 | (8,172) | 82.0% | 66,649 | 60,343 | (6,306) | 90.5% | 79,246 | 77,816 | (1,430) | 98.2% | 94,381 | 90,998 | (3,383) | 96.4% |
| 3 | General Practitioner (Generalist Doctor) | 14,049 | 6,173 | (7,876) | 43.9% | 21,923 | 9,027 | (12,895) | 41.2% | 26,560 | 11,003 | (15,557) | 41.4% | 32,199 | 12,369 | (19,830) | 38.4% |
| 4 | Laboratory Scientist | 4,581 | 1,355 | (3,226) | 29.6% | 8,151 | 2,991 | (5,160) | 36.7% | 11,125 | 4,470 | (6,655) | 40.2% | 14,933 | 5,806 | (9,126) | 38.9% |
| 5 | Laboratory Technician | 8,585 | 855 | (7,730) | 10.0% | 15,282 | 3,810 | (11,472) | 24.9% | 18,150 | 6,467 | (11,683) | 35.6% | 21,663 | 8,857 | (12,806) | 40.9% |
| 6 | Midwife | 14,002 | 12,786 | (1,216) | 91.3% | 17,586 | 24,756 | 7,170 | 140.8% | 22,440 | 32,688 | 10,248 | 145.7% | 28,805 | 37,944 | 9,139 | 131.7% |
| 7 | Nutritionist and Dietician | 5,937 | 334 | (5,603) | 5.6% | 6,038 | 2,254 | (3,784) | 37.3% | 6,445 | 3,981 | (2,464) | 61.8% | 7,094 | 5,534 | (1,560) | 78.0% |
| 8 | Pharmacist | 3,993 | 1,052 | (2,941) | 26.3% | 6,790 | 2,380 | (4,410) | 35.0% | 8,372 | 3,649 | (4,722) | 43.6% | 10,340 | 4,862 | (5,478) | 47.0% |
| 9 | Pharmacy Technician | 7,766 | 1,055 | (6,711) | 13.6% | 12,906 | 1,629 | (11,278) | 12.6% | 15,726 | 2,145 | (13,580) | 13.6% | 19,154 | 2,609 | (16,545) | 13.6% |
| 10 | Physician Assistant (Medical) | 8,590 | 3,118 | (5,472) | 36.3% | 14,511 | 5,305 | (9,206) | 36.6% | 17,633 | 7,292 | (10,341) | 41.4% | 21,487 | 9,097 | (12,390) | 42.3% |
| 11 | Registered General Nurse | 66,948 | 59,986 | (6,962) | 89.6% | 105,290 | 76,158 | (29,132) | 72.3% | 125,129 | 89,903 | (35,227) | 71.8% | 148,983 | 101,585 | (47,399) | 68.2% |
| | *Ghana* | *221,593* | *148,390* | *(73,203)* | *67.0%* | *337,392* | *225,454* | *(111,938)* | *66.8%* | *407,897* | *285,900* | *(121,997)* | *70.1%* | *495,273* | *333,770* | *(161,502)* | *67.4%* |

Note: Supply values for 2020 are not projections but the baseline data; 2020 values are the need-based baseline requirements

SAR = Staff Availability Ratio

by 2035. Similarly, the baseline supply of Pharmacy Technicians represents only 13.6% of the need-based requirement (which is expected to remain similar for the 15-year horizon of the projection), leaving an absolute need-based shortage of 6,711 in 2020, which is projected to increase dramatically to 16,545 by 2035. In the same vein, the baseline stock of Laboratory Technicians is estimated to represent only 10% of the need in 2020, leaving a need-based shortage of 7,730, which under the prevailing trends, the need-based shortage for Laboratory Technicians could reach 12,806 by 2035. However, in relative terms, the supply to need ratio is likely to improve gradually to 35.6% by 2030 and 40.9% by 2035.

Biomedical Scientists' baseline stock represents only 29.6% of the need-based requirement at the primary health care level, which is expected to improve marginally to 40.2% by 2030 and decline to 38.9% by 2035 if there are no interventions to influence the elements affecting production (and supply) and the need for Biomedical Scientists. The additional need of 3,226 Biomedical Scientists represents a 70.4% need-based shortfall in 2020, but the absolute shortage could escalate by 106.3% to 6,655 by 2030 if corrective interventions are not taken, and this shortage could reach even 9,126 by 2035. Similarly, Pharmacists' need-based shortage is estimated to be 2,941 at baseline in 2020, representing a 73.7% shortage of Pharmacists. Under the prevailing production and attrition rates, the absolute shortage could reach 5,478 by 2035, but relative to the need-based requirement, it will represent a 53% shortfall in 2035. Table 4 shows the projected supply versus need gap analysis in absolute and relative terms for all the health professionals considered in the analysis.

## Implications for planning the annual intake into health professions education institutions

Table 5 summarises the gaps and the recommended number of admissions per year for each category of health professionals to fill the projected need-based shortages. Notably, the analysis shows a need to ramp up most health professionals' training while maintaining or even reducing a few others, assuming the prevailing pass rates and attrition rates (as shown in Table 1) were held constant. For instance, 173 Pharmacy Technicians' annual intake needs to be

**Table 5. Need-based health professions training requirements.**

| No. | Health Workforce Category | Health Professionals Needed To Train | | | | | | |
|---|---|---|---|---|---|---|---|---|
| | | Need-based gap at baseline (2020) | Total additional production to fill need-based gaps by 2035 | Average changes in admissions required | Current annual admissions | Need-based optimal admissions per year | Need-based % change from current annual admissions | Suggested decision |
| 1 | Community Health Nurse | 17,293 | 42,125 | 3,033 | 4,184 | 7,217 | 72% | Increase |
| 2 | Enrolled Nurse | 8,172 | 3,383 | 257 | 8,379 | 8,636 | 3% | Increase |
| 3 | General Practitioner (Generalist Doctor) | 7,876 | 19,830 | 1,719 | 1,566 | 3,285 | 110% | Increase |
| 4 | Laboratory Scientist | 3,226 | 9,126 | 652 | 396 | 1,048 | 165% | Increase |
| 5 | Laboratory Technician | 7,730 | 12,806 | 913 | 682 | 1,595 | 134% | Increase |
| 6 | Midwife | 1,216 | (9,139) | (737) | 4,827 | 4,090 | -15% | Decrease |
| 7 | Nutritionist and Dietician | 5,603 | 1,560 | 109 | 429 | 538 | 25% | Increase |
| 8 | Pharmacist | 2,941 | 5,478 | 432 | 343 | 775 | 126% | Increase |
| 9 | Pharmacy Technician | 6,711 | 16,545 | 1,302 | 173 | 1,475 | 752% | Increase |
| 10 | Physician Assistant (Medical) | 5,472 | 12,390 | 987 | 638 | 1,625 | 155% | Increase |
| 11 | Registered General Nurse | 6,962 | 47,399 | 4,013 | 7,353 | 11,366 | 55% | Increase |
| | **Overall** | **73,203** | **161,502** | **12,680** | **28,970** | **41,650** | | |

tremendously escalated by almost 7.5-fold to about 1,475 annually. Also, the average annual intake of 4,180 for Community Health Nurses is projected to culminate in a future supply that will fail to offset the anticipated need-based shortage of 42,125 by 2035; hence there might be the need to increase their intake by some 72% to 7,217 per year. Similarly, General Practitioners annual enrolments need to be increased by 110% from 1,556 (inclusive of local and foreign training) to 3,285 annually.

In comparison, Biomedical Scientists and Physician Assistants may also need to be escalated by 165% and 155%, respectively. Thus, Biomedical Scientists' intake might need to increase from 396 to 1,048, and that of Physician Assistants might need to increase from 638 to 1,625 annually to address the anticipated need-based shortage by 2035. The annual intake of Pharmacists may also have to be expanded by 126% from 343 to about 775 per annum. The annual intake of Midwives could be scaled down by 15% from 4,827 (from direct and post-basic streams) to 4,090. On the other hand, Registered General Nurses may have to be ramped up by 55% from an annual average intake of 7,353 to 11,366 to meet the needs-based shortage by 2035, which are occasioned by expanding population health needs and attrition from general nursing to specialists and other careers as well as escalating out-migration.

## Cost implications for investing in health professions education to fill the need-based shortages

Using the average annual fee-paying rates and the estimated number of admissions required to fill the needs-based gaps by 2035 while adjusting for inflation (at a rate of 10% per annum), the

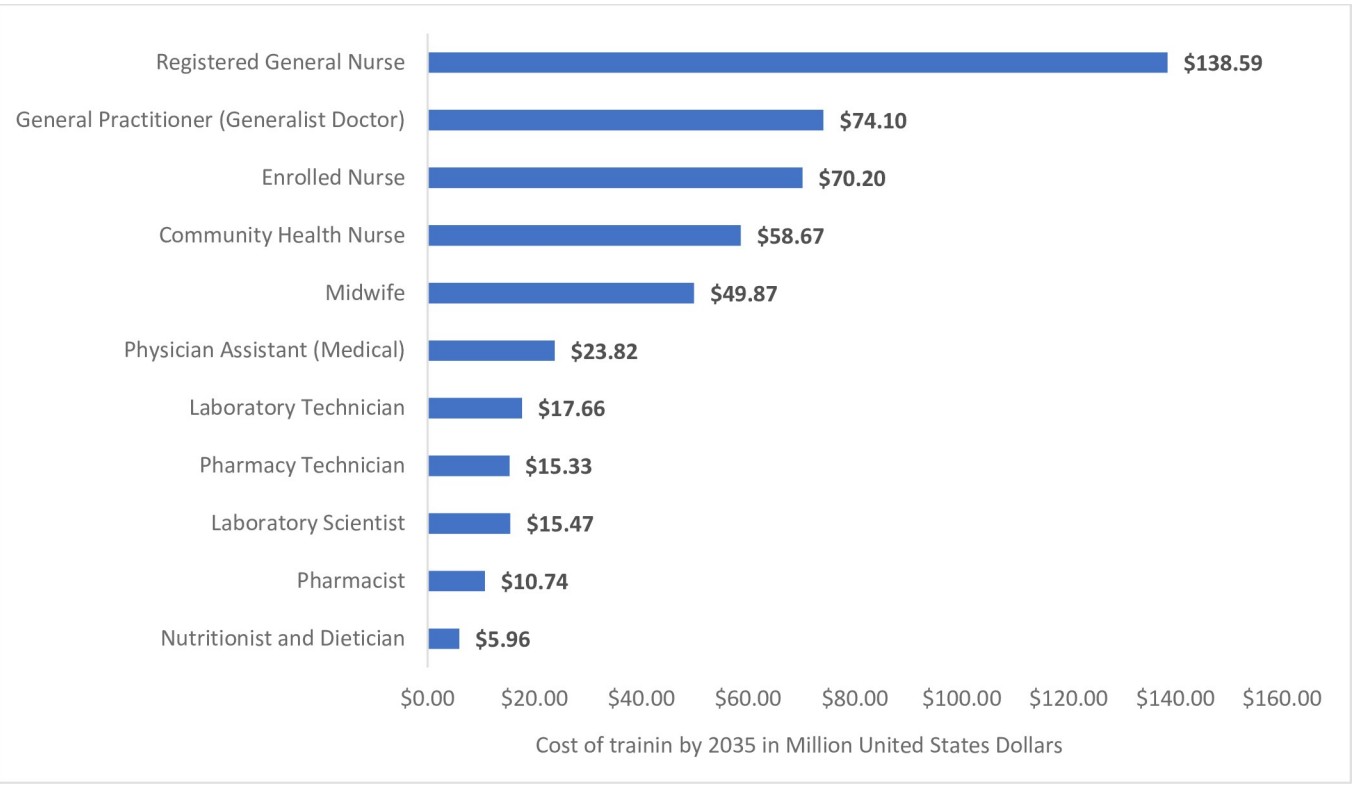

**Fig 3. Estimated cost for training health professionals needed to fill need-based shortages by 2035 (Million USD).**

cost of training to fill the aggregate gap of 170,529 health professionals across the 11 categories included in this analysis is estimated to be US$ 480.39 million. Of this amount, 28.8% (US$ 138.59 million) will be required for the training of Registered General Nurses up to 2035. Also, 15.4% (US$ 74.10 million) will be required for the training of General Practitioners. In comparison, Pharmacy Technicians who are expected to experience the most severe degree of shortage by 2035 will require US$ 15.33 million investments in their training (~3.2% of the aggregate estimated cost of training). Fig 3 shows the estimated cost of training for filling the anticipated needs-based gaps by 2035.

## Cost implications for employment planning for health professionals

Using the average income level of health professionals in the public sector (Table 1) and the projected supply and needs (Tables 2 and 3, respectively), we conservatively estimated the annual cost of employment with 10% annual salary upward adjustments or increases in response to inflation and based on historical patterns of annual wage adjustments. It is estimated that the cost of wages for the current stock of the eleven groups of health professionals (using public sector salary levels) was about US$667.24 million in 2020 which Registered General Nurses accounted for 44% of the aggregate cost (US$295.59 million) followed by Enrolled Nurses which accounted for 18% or US$117.06 million; General Practitioners and Community Health Nurses also accounting for 11% and 12% respectively. In contrast, the cost of employment to fill all the estimated need-based requirements in 2020 (assuming the supply was readily available to fill the need) was projected to be US$1.180 billion, of which 30% would have been for Registered General Nurses; 15% for General Practitioners; 12% for Community Health Nurses and 13% for Enrolled Nurses.

Holding all the assumptions mentioned above, by 2035, the projected cost of employment of the supply of these health professionals (i.e., sustaining the jobs of those already employed and recruitment of those being trained) would likely be around US$1.496 billion as compared to US$ 2.374 billion if the employment were to be based on the need. Thus, meeting the need-based requirements by 2035 would require an additional investment of US$ 878.37 million for employment compared with the cost of employment for the baseline supply in 2020. This implies that by 2035, US$ 2.374 billion must be planned for the employment of those that would have to be trained to fill the need-based shortages and for sustaining the employment of those currently available.

Without supply-side corrective measures to reflect the need-based gaps on the intake or number of admissions into health professions education institutions to correct the projected mismatches, the cost of inappropriate skill mix from the supply pipeline could be US$ 44.58 million by 2035 or 7% of the baseline cost of employing the available stock. Some of the mismatches could manifest in the form of unemployment and/or employment of skills that are not needed, thereby bloating the wage budget (especially in the public sector), and reducing the fiscal space for the employment of other equally essential health professionals. For instance, by 2035, the employment cost of the need-based requirement of Enrolled Nurses would be roughly 13% of the aggregate estimate (for the occupational groups considered in this projection), which would translate into US$297.14 million. However, the current supply trend would likely yield US$ 286.49 million, which represents 19% of the aggregate estimate if no corrective measures are taken in line with the need-based requirements. Similarly, whereas the cost of employing the needs-based requirement of midwives would be US$140.50 million (6% of the aggregate) by 2035, the cost of employment based on the prevailing trend of supply would likely be US$185.08 million or 12% of the aggregate estimate. Thus, failing to adjust the production of Midwives in line with the needs could cost US$44.58 million by 2030. In contrast, if no corrective measures are taken in increasing intake, the employment cost of the supply of General Practitioners in 2035 would be around US$145.41 million while the need require US$378.52—which will be US$233.11 million (or 62%) short of the cost of investment necessary to cover need-based employment of General Practitioners. Table 6 provides details of the projected cost of employment, comparing the prevailing supply trends and the need-based requirements for eleven health professionals.

## Sensitivity analysis: Testing the impact of the different assumptions on the projections

To examine the impact of key assumptions on the projection results, we conducted a one-way sensitivity analysis. We varied some of the assumptions one after the other, each time holding all others constant.

From our cross-sectional survey [48], the average practice variation in how much time health professionals perform the various task was ±18%. Altering all the professional standards by ±18%, the staffing requirements increased by 25% when the upper limits are assumed but reduced by 15% when the lower limits are assumed. This shows that the total variability in the staffing requirements attributed to practice variations is roughly 40%, which is substantial. It largely underscored that monitoring practice variations, especially those influenced by evolving technologies, are imperative for adjusting the staffing projections and, subsequently, the policies and strategies thereof.

One main distinction between this model and others is that the present model assumes that the future rate of change in disease pattern will mirror past trends, which may not necessarily be exact but somewhat overcomes a limitation where the previous models assumed that

**Table 6. Investments required for employment: Anticipated supply versus needs-based requirement of health professionals, 2020–2035.**

| NO. | HEALTH PROFESSIONAL | ESTIMATED EMPLOYMENT COST IN MILLION UNITED STATES DOLLARS (US$) | | | | | | | | PROPORTIONAL SHARE (%) OF THE ESTIMATED COST | | | |
|---|---|---|---|---|---|---|---|---|---|---|---|---|---|
| | | 2020 | 2025 | | 2030 | | 2035 | | | 2020 | | 2035 | |
| | | Needs-based | Supply-based | Needs-based | Supply-based | Needs-based | Supply-based | Needs-based | Supply-based | Needs-based | Supply-based | Needs-based | Supply-based |
| 1 | Community Health Nurse | 144.72 | 77.12 | 196.04 | 115.86 | 242.65 | 146.36 | 302.97 | 170.35 | 12% | 12% | 13% | 11% |
| 2 | Enrolled Nurse | 157.07 | 117.06 | 209.83 | 189.98 | 249.49 | 244.99 | 297.14 | 286.49 | 13% | 18% | 12% | 19% |
| 3 | General Practitioner (Generalist Doctor) | 181.67 | 72.57 | 257.72 | 106.12 | 312.22 | 129.34 | 378.52 | 145.41 | 15% | 11% | 16% | 10% |
| 4 | Laboratory Scientist | 31.12 | 8.37 | 50.34 | 18.47 | 68.71 | 27.61 | 92.23 | 35.86 | 3% | 1% | 4% | 2% |
| 5 | Laboratory Technician | 65.23 | 4.17 | 105.75 | 18.58 | 125.47 | 31.54 | 149.67 | 43.20 | 6% | 1% | 6% | 3% |
| 6 | Midwife | 75.15 | 62.37 | 85.80 | 120.75 | 109.47 | 159.44 | 140.52 | 185.08 | 6% | 9% | 6% | 12% |
| 7 | Nutritionist and Dietician | 40.34 | 2.06 | 37.29 | 13.92 | 39.81 | 24.59 | 43.81 | 34.18 | 3% | 0% | 2% | 2% |
| 8 | Pharmacist | 29.02 | 6.95 | 44.86 | 15.72 | 55.31 | 24.11 | 68.31 | 32.12 | 2% | 1% | 3% | 2% |
| 9 | Pharmacy Technician | 38.30 | 4.73 | 57.87 | 7.30 | 70.51 | 9.62 | 85.89 | 11.70 | 3% | 1% | 4% | 1% |
| 10 | Physician Assistant (Medical) | 58.36 | 19.26 | 89.62 | 32.76 | 108.90 | 45.04 | 132.71 | 56.19 | 5% | 3% | 5% | 4% |
| 11 | Registered General Nurse | 359.20 | 292.59 | 513.56 | 371.47 | 610.33 | 438.51 | 726.68 | 495.49 | 30% | 44% | 30% | 33% |
| *Overall* | | **1,160.99** | **667.24** | **1,617.45** | **1,010.95** | **1,955.92** | **1,281.14** | **2,374.44** | **1,496.07** | **100%** | **100%** | **100%** | **100%** |

Notes: Exchange rate used: 1 United States dollar = 5.8 Ghana Cedis; Estimates adjusted for wage increases which are usually benchmarked to the annual inflation rate, assumed to be about 10%.

present prevalence rates would remain constant into the future. When our assumption was relaxed, the need for health professionals was reduced by an average of 27%. In other words, in Ghana's context, future changes in disease patterns (if they mirror the previously observed trends) require about 27% more health workers within the next 15 years. Also, if the anticipated evolution in the disease burden were not taken into account, the health workforce requirements for the future would have been underestimated by some 27%. Fig 4 shows the trajectory of the projected aggregate needs-based requirements of the eleven groups of health professionals under various alternative assumptions.

From the supply side, if the pass rate of the various professional licensing examinations is increased to 100%, it would improve the overall supply of the eleven health professionals by 12% by 2035, which will, in turn, improve the overall supply to need adequacy ratio from 67% to 79% by 2030. A similar result was found if the overall attrition rates were reduced by 50%. Interventions to concurrently improve the pass rate to 100% and reduce the attrition rate by up to 50% could lead to a 25% improvement in the supply of health professionals by 2035, yielding a supply versus need adequacy ratio of 96%. Fig 5 shows the trajectory of eleven categories of health professionals under various alternative assumptions.

## Discussion

The analysis demonstrates the feasibility and value of using the need-based framework for health system-level planning for the health workforce, especially linking population health needs to quantifying the required intake into health professions education institutions. To the best of our knowledge, this study is the first attempt to undertake a multi-professional needs-

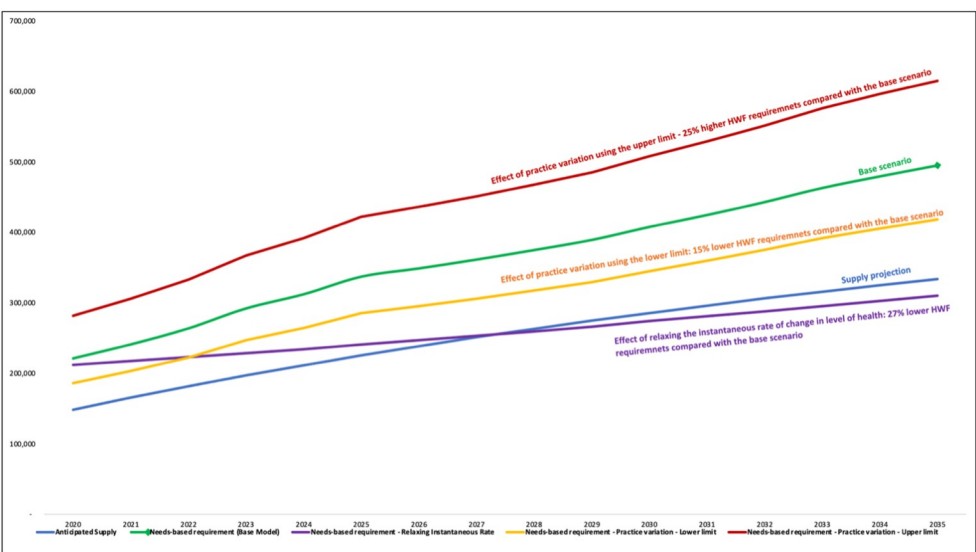

**Fig 4. Sensitivity analysis by varying need-related assumptions and variables.**

based health workforce projection in Ghana, which accounts for diseases and risk factors that constitutes 98% of the burden of morbidity and mortalities; taking a comprehensive approach (across public and private sectors) and for the level of health care that caters for 95% of health service utilisation. Previous works have either been based on health facility staffing norms linked to current workloads [13] or normative ratios focused on only Physicians, Nurses and Midwives [12] and/or limited the scope to only the public sector.

The projection shows that the health professionals' aggregate supply could meet 67% of the needs-based requirements, which is expected to remain somewhat similar, improving to 70% by 2030 and then declining back to 67.4% by 2035. Previous estimates [13], which were based on utilisation-oriented health facility staffing norms (health service development and analysis, HeSDA), suggested that, in aggregate, the public health sector had 68% of its requirements in

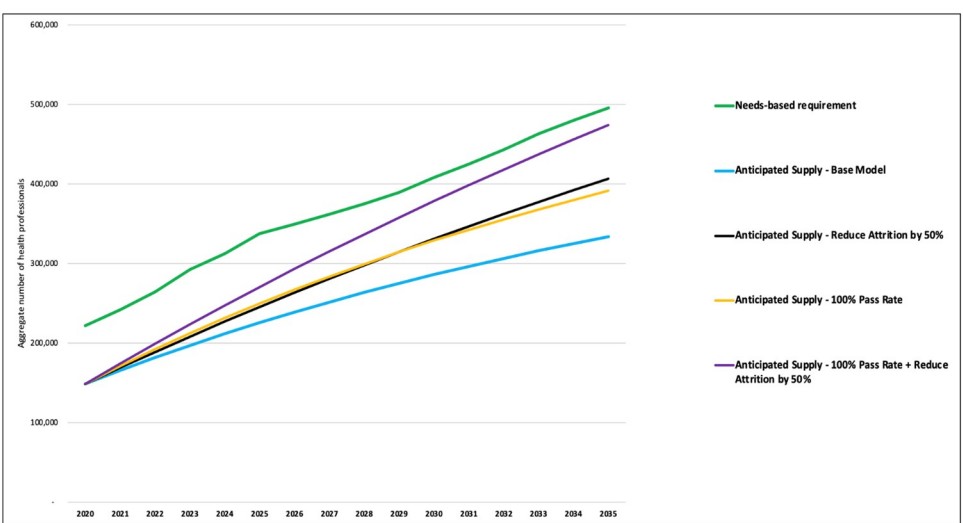

**Fig 5. One-way sensitivity analysis of supply-side variables.**

2016. A similar analysis conducted by the Ghana Health Service (the institution mandated to provide primary and secondary health services in Ghana) in using its health facility staffing norms versus those employed by 2018 concluded that it had 68.7% of the staff required for service delivery. However, when clinical staff alone was considered, it had only 49% of its requirements, which substantially differed from the 67% estimated in the present study. The present study also projects anticipated oversupply of midwives by roughly 32% by 2035 if interventions are not undertaken to ensure a balance.

Also, the supply of Enrolled Nurses and Registered General Nurses seem to be near equilibrium with the need at baseline. However, for Registered General Nurses, a widening shortage looms quickly the further into the future, the projection is extended. These should, however, be interpreted and compared with the previous works with caution as the present analysis took a comprehensive sector view, including public and private health services for primary health care. In contrast, the previous works were based on only publicly funded health facilities. Also, one comparative study found that needs-based models (as in the case of the present study) tended to produce about 53.5% (44% - 57%) higher staffing requirements as compared to utilisation or demand-based models [49] because the need-based model usually accounted for unexpressed or unmet health needs of the population which are missed out in utilisation or demand-based models. In line with this evidence, the current need-based projection has estimated HWF requirements for 2020 that are on average 42% (range: 26% - 63%) higher than that of the previous facility-based modelling study [13].

Nonetheless, the overarching findings of the present study are largely consistent with various reports and studies. For example, the annual holistic assessment reports of the Ministry of Health have since 2014 warned of over-supply of midwives, noting ". . . the productivity of midwives had significantly deteriorated by 51.3% . . . [resulting in] in a surplus of about 2,766 midwives [representing 28.9%] during the period under review [for 2017]" [19]. Although the Midwives' productivity analysis methodology used in the holistic assessments has been debated, a separate analysis also suggested that GHS in 2018 was 13% overstaffed with midwives albeit inequitably distributed [17], which could lend further credence to the potential midwives' over-production hypothesis. Similarly, the State of the World's Nursing in 2020 report also suggested a potential overproduction of nurses (professionals and auxiliaries combined) [50], while the State of the World's Midwifery Report 2021 estimated that Ghana had between 74% and 91% of its need for midwives. Whereas the aforementioned analyses adopted different methodologies from the present study and used global assumptions (instead of the country-specific data used in the present study), the similarities in conclusions tend to validate this study.

It is also important to note that since the present analysis took a comprehensive approach, the estimated needs and gaps comprise both the public and private sectors. However, translating the need-based requirements into actual demand or job creation to employ the health professionals, as well as adjusting the training outputs to respond to the needs, can take some time and depend on a web of multi-sectoral and multi-stakeholder actions–many of which may fall outside the remit of the health sector. From the employment perspective, there are bound to be a dynamic surplus of health professionals in which some health facilities may experience shortages and high vacancy rates for some health professionals even though they are available in the market for employment but rigidities in employment and budget allocation processes [14]. However, with the Government of Ghana's pledge to expand health infrastructure by establishing (and expanding) 111 district and regional hospitals, as well as infectious diseases and mental health hospitals, a programme that is known as 'Agenda 111' [51], the demand for health professionals, could move a bit closer to the need-based projections herein.

Also, from the supply side, although the analysis revealed a need for significant expansion in the intake of various health professionals while scaling down others, these have significant implications on infrastructure, equipment and faculty, which may take a longer time to address. For instance, Biomedical Scientists and Pharmacists' training institutions are rapidly introducing more extended periods of training (changing from 4 to 6 years), leading to doctoral qualifications, which could exacerbate infrastructural challenges as the students would have to spend two more years in school, thereby limiting the capacity for increased admissions. These may result in rigidities in adjusting intake into health professions education which could lead to dynamic shortage where the demand (jobs) may become available, but it takes several years to produce the needed calibre of health professionals. On the other hand, a dynamic surplus could be looming where the health professionals who are not in high demand would continue to be produced by the health professions education institutions either due to time lag in adjusting production downwards in response to the decreased demand or the health professions education institutions merely focusing on the income generation side especially in the profit-driven private-sector production. These institutions' focus may be to produce health professionals' categories requiring less input cost and rapid turnarounds, such as training enrolled nurses and top-up programmes. As some of these programmes have lower entry requirements, the market for health professionals education (applicants) abounds, but soon after employment, the graduates seek career advancements through top-up courses to become Midwives or Registered General Nurses [14].

The analysis shows that without corrective interventions to reflect the projected needs-based gaps on the intake (number of annual admissions) into health professions education institutions, the cost of inappropriate skill mix could reach US$44.58 million or 7% of the cost of employing the baseline supply of the health professionals, mainly from the potential over-supply of midwives. This will likely manifest in the form of unemployment and/or employment of skills not needed, which will balloon the wage budget (especially in the public sector), reducing the fiscal space for the other equally essential health professionals with high vacancy rates. A recent analysis that supports this finding revealed that about 28% (range:16–38%) of public health sector wage was already being lost to inefficiency due to maldistribution and inappropriate skill mix, which costs the government some US$295.4 million annually [52].

Finally, as the estimates of needs-based requirements reported in this paper are aggregates, they do not guarantee geographical distributional equity but may be best suited for sector-wide policies and planning regarding targets for training and employment while other evidence-based management tools such as flexible staffing norms or WISN analysis is used to ensure equitable distribution of the health professionals produced. The analysis does not include the disaggregation of how much of the need-based requirements could be absorbed by the public or private sectors. This study was focused on needs and gaps for training rather than a demand-based labour market analysis which a separate study in Ghana is imperative.

## Limitations of the study

The study has a number of limitations relating to the scope, data quality and methodological assumptions that should be taken into account when interpreting the findings and/or using them for decision making.

First of all, whereas the model could be applied for the entire health service need for health professionals in Ghana (from primary to tertiary and quaternary settings of health care), the present analysis focused on primary health care, which is accessed by more than 95% of the population during outpatient care. Primary health care currently accounts for at least 62.5% of health workers employed in the public sector [53]. Hence, when interpreting the findings, it

must be borne in mind that it may not represent the complete picture of need-based requirements of some of the health professionals that may also be needed at the more sophisticated secondary, tertiary and quaternary levels of care (these levels combine to roughly provide advanced services for 5% of the population's health needs).

Secondly, the input data into the model were triangulated from various sources with varying quality and completeness levels. For instance, although it would have been preferable to rely solely on periodic national surveys for disease prevalence (and incidence) data, there was no 'STEPwise approach to non-communicable disease surveillance' (STEPS) survey done in Ghana to reliably obtain the prevalence of most non-communicable diseases and their risk factors. Under the circumstance, peer-reviewed papers were used in which we prioritised systematic reviews, single large-scale surveys and other well-conducted analytical pieces. Similarly, we relied on health professional regulatory bodies' registers for data on health professionals' stocks. However, due to weak regulatory enforcement, it is possible that some health professionals may not be up-to-date in renewing their practice licenses and hence may have been missed, thereby underestimating the stock of health professionals; or on the other hand, those that have migrated abroad or not practising their professions may remain in good standing with the regulatory bodies and thereby inflating the estimated stock. Also, we used data from the public sector (which employs about 80% of the health professionals) relating to average income and attrition rates, but the income and attrition in the public sector could vary from that of the private sector. Future studies to estimate and compare income and attrition rates between public and private sectors would help refine the projections.

Thirdly, methodologically, the model makes an explicit assumption that the current prevalence rates of diseases and their risk factors or even coverage rates will not remain constant throughout the horizon of the projection, but data on what to expect in the future is often not available for all diseases and risk factors. Hence, the model assumes that the future rate of change in disease (and risk factors) pattern will mirror past trends. Although this may seem a hard assumption, we believe that its benefit in the approximation outweighs the limitation where previous models assumed that present prevalence rates would remain constant into the future. Nonetheless, the absence of independent projections of the future trajectories of the prevalence of various diseases and risk factors for the country do expose the projections herein to potential deviations. However, in the sensitivity analysis, we demonstrate that relaxing the aforesaid methodological assumption (assuming the constant level of health as done in previous models) could underestimate the aggregate staffing needs by 27%. We thus, lean on the argument of leading need-based theorists that *"problems with data are not avoided by adopting or reverting to the conceptually invalid models most commonly used by HRH planners . . . [but]. . . the continual refinement of the application of a conceptually valid approach is superior to adopting conceptually invalid approaches based on the availability of data"* [26].

Finally, although health emergencies, such as the COVID-19 pandemic, have a significant implication on health professionals' requirements and hence should be taken into account in need-based projections, there was limited information at the time of modelling if the COVID-19 will become endemic like other infectious diseases or will be eliminated within the very near future. Added to that, there was no population-based prevalence data in Ghana; hence, it was not feasible to include it in future projections. There are appropriate tools for estimating health workforce requirements in situations of health emergencies and disrupted health systems.

## Conclusion

The study demonstrates the feasibility of using the need-based framework for national-level planning, which we have included a fully functional tool developed in Microsoft Excel as S1

File for possible use in other contexts. It shows potential value in linking population health needs to the required intake of students into health professions education institutions. It reveals that Ghana's supply in 2020 satisfies about 67% of the aggregate need-based requirements for primary healthcare for the eleven categories of health professionals, but a gap of 33% (or roughly 73,203) remains. Without any corrective intervention, the aggregate needs-based shortage in supply will likely be 161,502 by 2035, with the supply of 6 out of the 11 health professionals (~54.5%) failing to match 50% of the needs by 2035, but that of Midwives will likely be oversupplied by 32% in 2035. Priority areas for health professions education include scaling up the production of Pharmacy Technicians by 7.5-fold; General Practitioners by 110% whilst scaling down Midwives production by 15%. About US$ 480.39 million investment is required in health professions education to fill the need-based gaps and correct the mismatches by 2035, without which there will be a 33% shortage of essential health professionals and coupled with at least US$44.58 million annual losses to inappropriate skill mix. The adverse technical impact on interprofessional team compositions and, eventually, health care quality cannot be overemphasised. Finally, it is recommended that the analysis is updated regularly especially after every major national health survey, when significant technological or clinical practice changes are adopted, when scope of practice for health professionals evolve, when significant curriculum changes are made, or when important population health events occur, such as epidemics, disasters etc.

## Supporting information

**S1 Table. Summary table on level of health evidence (prevalence of diseases and risk factors).**
(DOCX)

**S1 File. Population needs-based simulation model for health workforce planning (Microsoft Excel-based model).**
(XLSM)

## Acknowledgments

We would like to Hamza Ismaila, Jerry Kwame Asamoah and Richmond Sowah for their support in retrieving secondary data from the archives of Human Resource Division of the Ghana Health Service. Dr Yolande Heymans and Mrs Paula Jardim are acknowledged for their respective technical and administrative support. Participants of the NWU CHPE seminar on "Predicting Primary Healthcare Workforce Needs Towards Health Professions Education and Employment Investment Planning: A Mathematical Model" are greatly acknowledged for their valuable feedback.

## Author Contributions

**Conceptualization:** James Avoka Asamani, Christmal Dela Christmals, Gerda Marie Reitsma.

**Data curation:** James Avoka Asamani.

**Formal analysis:** James Avoka Asamani.

**Funding acquisition:** James Avoka Asamani.

**Methodology:** James Avoka Asamani, Gerda Marie Reitsma.

**Software:** James Avoka Asamani.

**Supervision:** Christmal Dela Christmals, Gerda Marie Reitsma.

**Validation:** Christmal Dela Christmals, Gerda Marie Reitsma.

**Writing – original draft:** James Avoka Asamani.

**Writing – review & editing:** James Avoka Asamani, Christmal Dela Christmals, Gerda Marie Reitsma.

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
