## [Decision Letter · Decision Letter 0]

11 Aug 2021

PONE-D-21-05604

Modelling the supply and need for health professionals for primary health care in Ghana: Implications for health professions education and employment planning

PLOS ONE

Dear Dr. Christmals,

Thank you for submitting your manuscript to PLOS ONE. After careful consideration, we feel that it has merit but does not fully meet PLOS ONE’s publication criteria as it currently stands. Therefore, we invite you to submit a revised version of the manuscript that addresses the points raised during the review process.

The reviewers considered this a well-written document on the whole with facts supporting the conclusions. A few minor revisions are recommended including fixing the references and some minor typographical errors. In addition, 1 reviewer has recommended presenting a consistent data set in the same order with which I also agree.

We look forward to receiving your revised manuscript.

Kind regards,

Elizabeth S. Mayne, M.D.

Academic Editor

PLOS ONE

Additional Editor Comments (if provided):

The reviewers considered this a well-written document on the whole with facts supporting the conclusions. A few minor revisions are recommended including fixing the references and some minor typographical errors. In addition, 1 reviewer has recommended presenting a consistent data set in the same order with which I also agree.

Journal Requirements:

Reviewers' comments:

Reviewer's Responses to Questions

**Comments to the Author**

1. Is the manuscript technically sound, and do the data support the conclusions?

Reviewer #1: Yes

Reviewer #2: Yes

2. Has the statistical analysis been performed appropriately and rigorously? 

Reviewer #1: Yes

Reviewer #2: Yes

3. Have the authors made all data underlying the findings in their manuscript fully available?

Reviewer #1: Yes

Reviewer #2: Yes

4. Is the manuscript presented in an intelligible fashion and written in standard English?

Reviewer #1: Yes

Reviewer #2: Yes

5. Review Comments to the Author

Reviewer #1: Congratulations for tackling a difficult but extremely relevant subject. Planning for healthcare needs, if done properly offers all stakeholders an objective tool on which to base key decisions including resource allocation. The human resource is the most important component of such plans.

The article conforms with Plos One submission guidelines.

The writing style, grammar, spelling and formatting are all exceptionally well done, save for a few minor errors.

Figures and illustrations are appropriate and relevant to the points advanced in the text.

This reviewer did not have the capacity to confirm the veracity and assumptions behind the extensively used mathematical formulae. These formulae are the single most important tool used to arrive at the various conclusions made. However, the notes and explanations were reviewed in-depth and each equation was interrogated. The equations are easy to understand and logical. This aspect of the research therefore stands out as being particularly innovative.

The Findings (Results) are well presented and the graphics are in-depth and relevant.

The Discussion is logically presented and focused.

This reviewer is concerned about the disease spectrum's contribution to health needs. Projecting the disease spectrum in 2035 cannot be an exact science as several variables cannot be forecast accurately. This uncertainty has an impact on several other parameters such as the type of Healthcare Professional that will be required and the cost of addressing the specific disease challenges. Mass human migration into Ghana for instance can steeply increase the needs as can, any sustained outbreak of infectious disease. Furthermore, unnatural healthcare challenges such as disasters, war and casualties are not factored into the mathematical modelling. To their credit, the authors deal with this weakness under "Limitations".

The discussion on the type of healthcare professional needed, is weakened by the fact that some assumptions made are speculative. For instance, the scope of practice of the Biomedical Scientist vs Laboratory Technician which is based on Ghana Government (Employer) publications may not be factual. This is an important factor to consider in needs analysis.

The Limitations are in-depth and mostly factual (partially discussed above).

The Conclusion is well structured.

Overall, this is an excellent work requiring no specific alteration.

Reviewer #2: The manuscript appears to be technically sound and the data support the conclusion. Data have been validated and a sensitivity analysis performed. All data underlying the findings of the manuscript are fully available. The statistical analysis is a function of the modelling software used in the study.

The ethical aspects of the study have been complied with in that institutional ethic approval has been obtained from both the North West University and Ghana Health service Ethics Committee.

Specific comments and recommendations have been highlighted in the attachment.

6. PLOS authors have the option to publish the peer review history of their article (what does this mean?). If published, this will include your full peer review and any attached files.

Reviewer #1: No

Reviewer #2: No

---

## [Author Response · Author response to Decision Letter 0]

16 Aug 2021

We very thankful to the esteemed reviewers and editor for the positive feedback, suggestions and guidance. Your valuable comments have helped us to revise and improve the manuscript. We have attached a point-by-point response and/or explanation the comments for your kind consideration. We are looking forward to hearing from you soon.

---

## [Decision Letter · Decision Letter 1]

15 Sep 2021

Modelling the supply and need for health professionals for primary health care in Ghana: Implications for health professions education and employment planning

PONE-D-21-05604R1

Dear Dr. Christmals,

We’re pleased to inform you that your manuscript has been judged scientifically suitable for publication and will be formally accepted for publication once it meets all outstanding technical requirements.

Kind regards,

Elizabeth S. Mayne, M.D.

Academic Editor

PLOS ONE

Additional Editor Comments (optional):

Reviewers' comments:

Reviewer's Responses to Questions

**Comments to the Author**

1. If the authors have adequately addressed your comments raised in a previous round of review and you feel that this manuscript is now acceptable for publication, you may indicate that here to bypass the “Comments to the Author” section, enter your conflict of interest statement in the “Confidential to Editor” section, and submit your "Accept" recommendation.

Reviewer #1: All comments have been addressed

Reviewer #2: All comments have been addressed

2. Is the manuscript technically sound, and do the data support the conclusions?

Reviewer #1: Yes

Reviewer #2: Yes

3. Has the statistical analysis been performed appropriately and rigorously? 

Reviewer #1: Yes

Reviewer #2: Yes

4. Have the authors made all data underlying the findings in their manuscript fully available?

Reviewer #1: Yes

Reviewer #2: Yes

5. Is the manuscript presented in an intelligible fashion and written in standard English?

Reviewer #1: Yes

Reviewer #2: Yes

6. Review Comments to the Author

Reviewer #1: No new comments. Congratulations on this paper which addresses a very important aspect of healthcare delivery.

Reviewer #2: Dear Author please note the following for correction.

1. Line 413 reads levers of production instead of levels of production.

2. References 48 and 49 refers to the same publication.

7. PLOS authors have the option to publish the peer review history of their article (what does this mean?). If published, this will include your full peer review and any attached files.

Reviewer #1: No

Reviewer #2: No